# Background and Blended Spectral Line Reduction in Precision Spectroscopy of EUV and X-ray Transitions in Highly Charged Ions

Adam Hosier [1,*], Dipti [2,†], Yang Yang [1], Paul Szypryt [3], Grant P. Mondeel [1], Aung Naing [2], Joseph N. Tan [4], Roshani Silwal [5], Galen O'Neil [3], Alain Lapierre [6], Steven A. Blundell [7], John D. Gillaspy [2,8], Gerald Gwinner [9], Antonio C. C. Villari [6], Yuri Ralchenko [4] and Endre Takacs [1,2]

[1] Department of Physics and Astronomy, Clemson University, Clemson, SC 29634, USA
[2] Associate, National Institute of Standards and Technology, Gaithersburg, MD 20899, USA
[3] National Institute of Standards and Technology, Boulder, CO 80305, USA
[4] National Institute of Standards and Technology, Gaithersburg, MD 20899, USA
[5] Department of Physics and Astronomy, Appalachian State University, Boone, NC 28608, USA
[6] Facility for Rare Isotope Beams, 640 S Shaw Ln, East Lansing, MI 48824, USA
[7] CEA, CNRS, IRIG, SyMMES, University of Grenoble Alpes, 38000 Grenoble, France
[8] National Science Foundation, Alexandria, VA 22314, USA
[9] Department of Physics and Astronomy, University of Manitoba, Winnipeg, MB R3T 2N2, Canada
* Correspondence: ahosier@g.clemson.edu
† Present address: International Atomic Energy Agency, A-1400 Vienna, Austria.

**Abstract:** Extreme ultraviolet spectra of Na-like and Mg-like Os and Ir were recorded at the National Institute of Standards and Technology using a grazing incidence spectrometer. We report a method in EBIT spectral analysis that reduces signals from contaminant lines of known or unknown origin. We utilize similar ion charge distributions of heavy highly charged ions that create similar potentials for lighter contaminating background elements. First-order approximations to ion distributions are presented to demonstrate differences between impurity elements with and without heavy ions present.

**Keywords:** spectroscopy; highly-charged ions; EBIT; line blend

## 1. Introduction

Over the past few decades, the Electron Beam Ion Trap (EBIT) has become a popular and widely used instrument in the generation and trapping of highly charged ions. There has been recent excitement surrounding the study of highly charged ions (HCIs) due to a plethora of reasons, some examples include HCIs being potential candidates for the testing of fundamental symmetries and QED in a new era of precision measurements [1,2], the study of nuclear properties taking advantage of the compressed wavefunctions of HCI [3,4], and in general plasmas being the dominant state of matter in the early and current universe [5] To study these exotic atomic systems, the desired charge state ions themselves need to be generated and trapped in the EBIT. In this paper, we investigate a general problem in precision EUV and X-ray spectroscopy of HCI: the reduction of contaminating features affecting spectral analysis.

## 2. Spectroscopy in the EBIT

As an example for precision spectroscopy and the main device for our EUV and X-ray studies, the NIST EBIT has several different spectrometers attached to the system. The main instruments are currently the transition-edge sensor (TES) microcalorimeter [6] and the extreme ultraviolet (EUV) flat field grazing spectrometer with a charged-coupled device (CCD) detector [7]. The detectors ultimately produce one-dimensional spectra that give the

number of photons in a particular wavelength range. The spectrometers utilize wavelength values and uncertainties of well-known transitions of various injected and contaminant elements (neon, oxygen, iron, xenon, barium, and others) in order to provide an absolute wavelength calibration for the devices. To reduce electronic noise in spectra, the EUV spectrometer CCD detector is liquid nitrogen cooled.

In order to precisely measure the wavelength of a particular transition, there needs to be a favorable combination of a good signal-to-noise ratio of the measured transition in conjunction with a relatively low calibration uncertainty at the location of the line. In addition, it is even more important for the measured transition to be isolated from other features so as to not affect the position and shape of the spectral line. Assuming that the natural line broadening of the line of interest is negligible compared to instrumental broadening, the fitting of a Gaussian function can be used to determine the parameters of the spectral line.

When the composition of the ion cloud is known, theoretical collisional radiative modeling software [8] can predict line intensities emitted by the plasma, receiving inputs from precise atomic structure calculations [9,10]. However, if lines from known or unknown origin are present and affecting the lineshape of the measured line or spectral features used for calibration, this will ultimately reduce the accuracy of the measurement.

## 3. Ion Cloud Confinement

The electron beam is the main driver for all of the physics inside of the EBIT and is used to generate and trap the desired ion state. The capability of producing a diverse range of charge states in the EBIT is the result of four main processes: electron impact ionization, dielectronic recombination, radiative recombination, and charge exchange. The variable energy and high current of the compressed beam of electrons allows species to be ionized to charge states of interest, which are then radially trapped through the space charge of the electron beam itself.

The axial trapping is achieved by producing electric fields from a combination of three drift tubes with selective voltages to generate a potential well for the ions to reside in. Another advantage of the adjustable voltage drift tubes is their capability to occasionally force the ions out of the trap axially to allow newly generated ions to enter the trap. A superconducting magnet compresses the beam of electrons to a current density of the order of 1000 A/cm$^2$ [11], while electrostatic potential differences accelerate the electron beam energy upwards of 30 keV [12]. A magnet coil nullifies the magnetic field near the cathode end of the electron gun to allow high compression in the region of the superconducting magnet.

The range of charge states that are present in the trap at any given time (with the exception of the process of dumping the ions with drift tube voltage switches) and the distribution of the charge states across the profile beam are dictated by the interaction between the electron beam and the ions. This electrostatic interaction results in the higher charge states residing towards the center of the trap and the distribution of lower charge states extending over the regions further away from the electron beam and ultimately even escaping the trap, thereby, providing evaporate cooling for the remaining ion cloud [12,13]. In thermal equilibrium, we can assume that the relative populations of these various charge states follow the Boltzmann distribution [14]:

$$n_i(\rho) \propto exp(-\frac{q_i(V_e(\rho) + V_c(\rho))}{k_B T}) \tag{1}$$

where $q_i$ is the charge of ion species $i$, and $V_e(\rho)$ and $V_c(\rho)$ are the potentials due to the electron beam and the ion cloud, respectively, at some distance $\rho$ from the beam axis. $T$ is the characteristic temperature of the captured ions. The potentials to first approximation can

be considered to be resulting from distributions of negative and positive charges with radial Gaussian density dependence and linear charge densities of $\lambda_e$ and $\lambda_c$ as shown in [14]:

$$V_j(\rho) \simeq V_{0,j} \begin{cases} 2ln(\frac{\rho}{a_j}) + 1 & \rho > a_j \\ \left(\frac{\rho}{a_j}\right)^2 & \rho \leq a_j \end{cases} \tag{2}$$

$$V_{0,j} = 1.08 \frac{\lambda_j}{4\pi\epsilon_0}. \tag{3}$$

Here, $j$ is either $e$ or $c$ for the electron beam or the ion cloud, respectively. The scaling parameters $a_j$ are the radii of the electron beam and the ion cloud defined above as Gaussians.

### 4. Removal of Blended Spectral Lines

Dealing with blended lines in spectra is generally challenging especially if their origin is unknown. Knowledge about the EBIT ion cloud, however, can offer some insight into a possible handling of the problem. This issue arose in our recent experiment when the wavelengths of Na-like and Mg-like Os and Ir lines were measured with high statistical significance in the EUV (Figure 1) range of the spectrum.

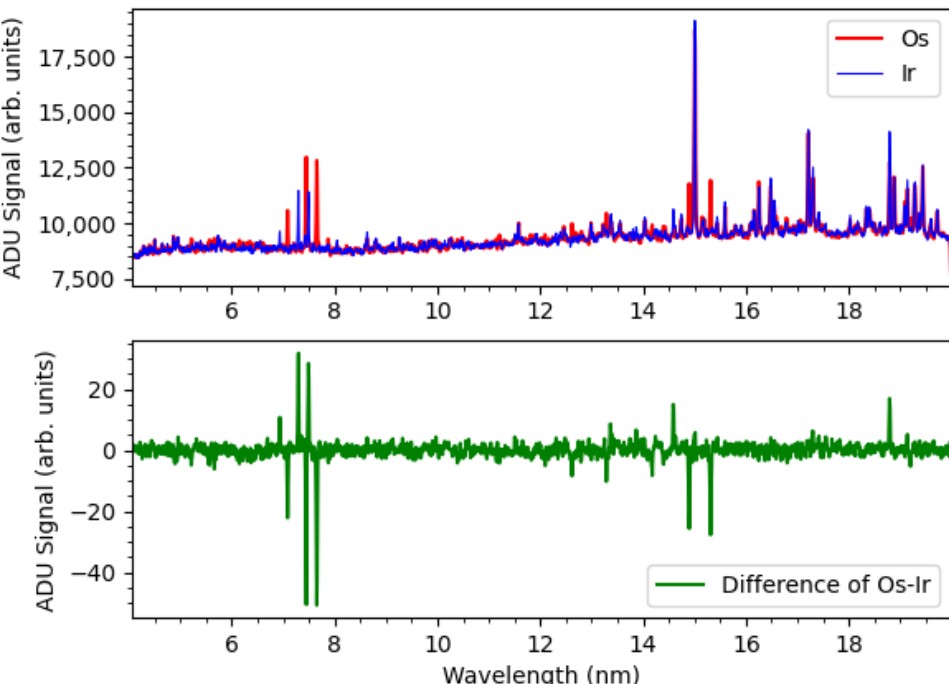

**Figure 1.** (**Top**) Spectra of highly charged Os and Ir in the EUV range at 18.1 keV electron beam energy and 150 mA electron beam current, exhibiting features from both elements and features that are present in both spectra. (**Bottom**) Subtraction method applied to above time-normalized spectra.

Our method to eliminate contaminant lines in the spectra involves taking the difference between two time-normalized spectra that are produced from very similar EBIT conditions. The main constraint with the application of this proposed method is that it requires the species introduced to the EBIT to not differ by a large atomic number or effective charge between the two species.

This typically holds true for sufficiently heavy, highly charged nuclei as the change in the mass or charge between two species is nearly negligible over a small increase in atomic number. Additionally, the EBIT parameters similar to the example given in Table 1 should be consistent between the two trapped species.

**Table 1.** Typical values of EBIT operating parameters at the NIST EBIT.

| Parameter | Value |
|---|---|
| Electron Beam Current | 10 mA to 150 mA |
| Electron Beam Energy | 1.0 kV to 30.0 kV |
| Dump Cycle | few s |
| Pressure of Gas Injector | $3 \times 10^{-3}$ Pa |
| Visible Trap Length | 2 cm |
| Magnetic Field | 2.7 T |

The application of this new method results in time-normalized subtracted spectra, which exhibit spectral features unique to the heavy elements injected into the EBIT while eliminating signals from contaminant ions that were present in the EBIT at the time both spectra were taken.

The rationale in using this *subtraction method* results from the nearly identical electric potentials that background light elements effectively see while they occupy the region of the electron beam from which their radiation originates. As of the close similarity of the heavy ions, the charge states and spatial distributions of the contaminant elements are nearly identical (see Equation (1)), thereby, producing the same spectral response from these ions.

If there is a blend with a measured line, this method can allow one to remove the blend, given it is from a contaminant element that was present in the EBIT during the injection of both of the heavy elements. To showcase this method, an example is the spectroscopic data of highly charged osmium and iridium ions, observing the Na-like $3s - 3p_{1/2}$ and the Mg-like $3s^2 - 3s3p_{1/2}$ transitions in the EUV regime (approximately 7.2–7.7 nm).

The above plots in Figure 1 show how the reduction background signal lines that are not from either Os or Ir are eliminated almost entirely, and we are left with spectra that are only associated with Os and Ir in Figure 2 (one must be mindful of the difference in acquisition times between spectra and apply this method to time-normalized spectra as seen in the bottom plot of Figure 1).

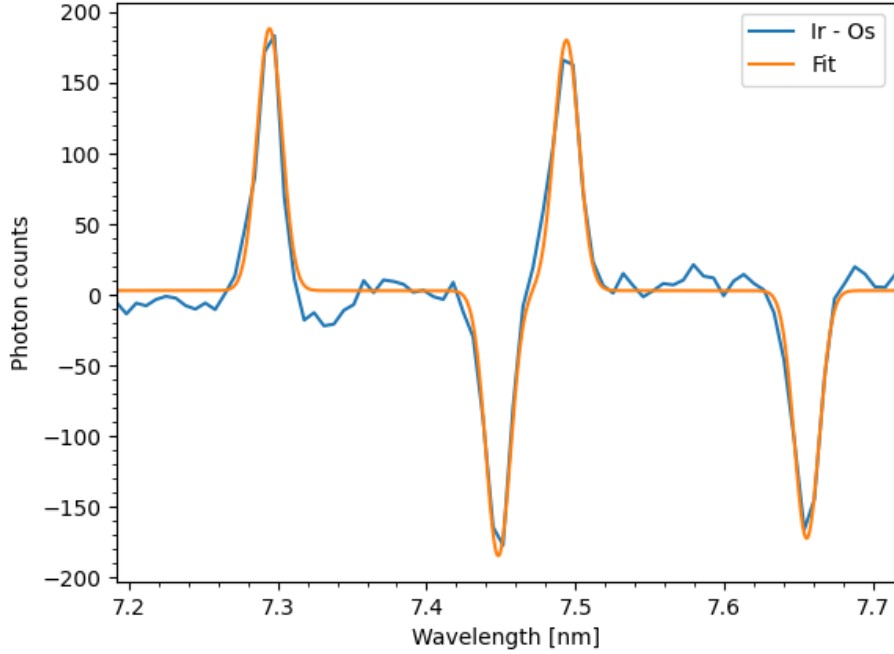

**Figure 2.** Subtracted spectra between the two spectra given in Figure 1 fitted with a sum of four Gaussian functions plus a constant offset.

### 5. Ion Distributions

Aside from empirical evidence found in the experimental spectra, similarities in charge-state distributions can be found from Equation (1), which describes the population distributions in the EBIT radial potential. Utilizing these expressions, one can generate the radial Boltzmann distributions of the ion cloud through first-order approximations for high and low Z elements assuming similar electron beam neutralization at identical EBIT operating parameters. Prior to creating distributions, the correct effective potentials generated by the electron beam and ion cloud must be generated.

One can observe an increase in the summed potential with an increase in electron beam neutralization, thus, verifying whether the potentials are correct (Figure 3). From here, the Boltzmann distributions for various scenarios can be calculated to a first-order approximation including both the potential of the electron beam and potential due to heavy ions for a distribution with respect to a lighter charged ion in addition to the presence of only a light or heavy ion. To demonstrate this idea, let us use an arbitrary light ion charge of $Q = 10^+$ as seen in Figure 4.

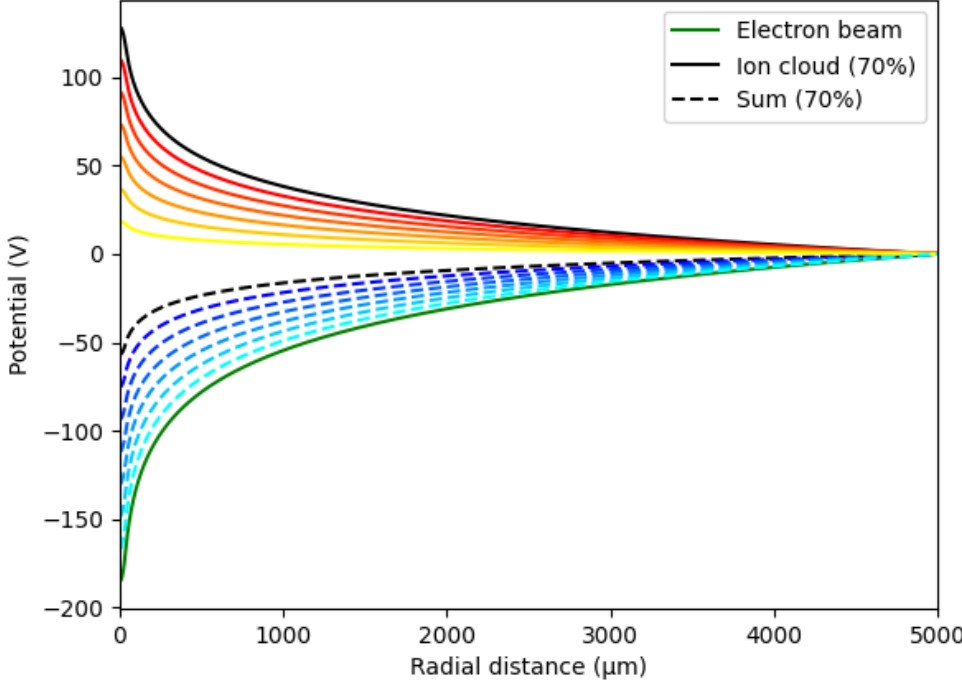

**Figure 3.** Calculated potential for a heavy ion cloud ($Q = 65^+$, solid line) and the total potential including an ion cloud and electron beam (dotted line) for various neutralization conditions ranging from 10% (yellow and light blue) to 70% (dark red and dark blue) electron beam neutralization for a beam energy of 18 keV, beam current of 150 mA, and ion cloud temperature of 20 MK.

From visual inspection of Figure 4, the comparison between the distributions of $Q = 10^+$ and heavy ions ($65^+$) and $Q = 10^+$ and heavy ions ($66^+$) results in similar radial distributions for light background ions, giving similar background charge-state distributions. The average electron density is of interest since the overlap of the electron distribution with the extent of the ion cloud determines the interactions the ions have with the plasma. The average electron densities for only a light ion of $Q = 10^+$ are $1.29 \times 10^{11}$ $e^-/cm^3$.

The average electron densities for the distributions of $Q = 10^+$ with heavy ions ($65^+$ and $66^+$) are $6.25 \times 10^{10}$ $e^-/cm^3$ and $6.41 \times 10^{10}$ $e^-/cm^3$, respectively, while the average electron density of the electron beam is $3.05 \times 10^{12}$ $e^-/cm^3$. As noted before, this is only a first-order approximation as the true potential is dictated by Poisson's equation and Gauss' law along with Dirichlet boundary conditions (drift tube voltages) to, thus, provide a self-consistent numerical solution potential for the electron beam and ion cloud.

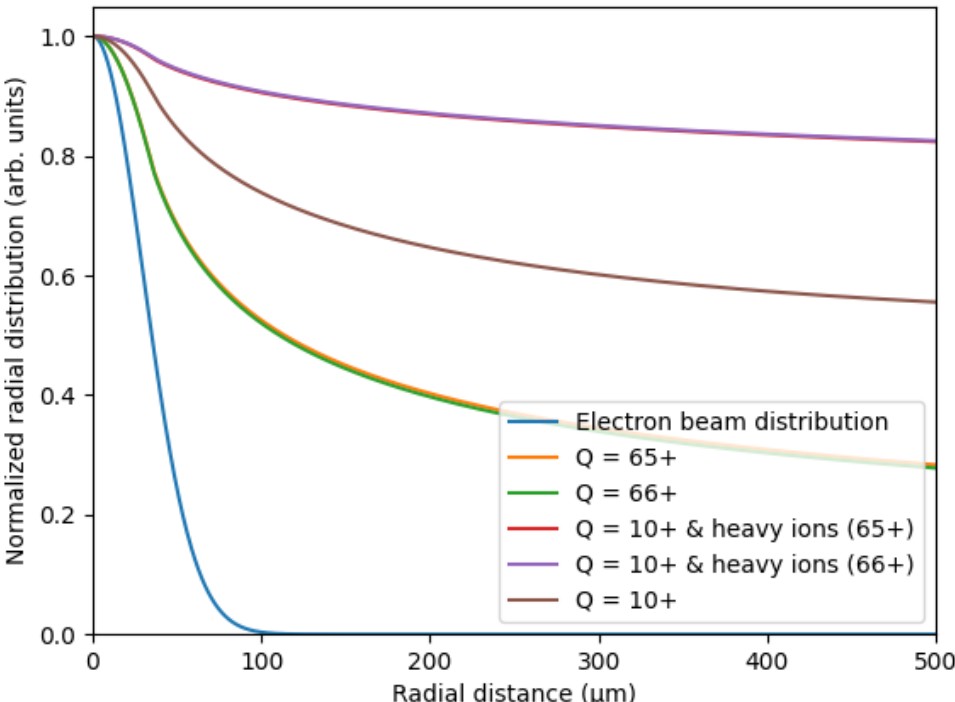

**Figure 4.** Boltzmann distributions of ions with charge $Q = 10^+$, $65^+$, $66^+$, $10^+$ with $65^+$ present, and $10^+$ with $66^+$ present in identical electron beam ion trap conditions (18 keV beam energy, 150 mA beam current, and 70% neutralization with all ions in thermal equilibrium at a temperature of 20 MK).

## 6. Discussion and Conclusions

The limitations for this methodology depend on the overlap of the ion cloud distribution with the extent of the electron beam. To quantify this, one should evaluate the effective electron density that the light ion cloud sees in the presence of heavier elements. The difference in the relative electron density should be negligible in order to subtract one spectra from another.

This new methodology has proven useful in spectra of highly charged Os and Ir with Na-like and Mg-like ($65^+$ and $66^+$) charge states where unknown spectral lines might have been blended with the Na-like and Mg-like lines in the EUV regime. Typical reduction techniques in the blended signal through comparison of FWHM of the spectral line with other nearby lines and wavelengths through precise atomic structure calculations are useful tools but occasionally fail when insufficient information is available.

This technique can be applied generally with any high-temperature plasma system where numerous atomic species are trapped in a similar fashion, given that the user correctly evaluates the extent of the respective Boltzmann distributions in the trapping region for a particular set of species. The development of remedies for these types of issues is necessary to reduce the uncertainties in precision measurements that are pushing the limits of QED and the standard model. Particularly in low-density plasmas, such as in the EBIT, the method that we detail here can be a useful tool to reduce the influence of systematic shifts due to backgrounds and line blends down to negligible levels; however, it cannot be assumed to eliminate them entirely.

**Author Contributions:** Conceptualization, A.H., E.T. and Y.R.; Investigation, A.H., D., Y.Y., P.S., G.P.M., A.N., J.N.T., R.S., G.O., A.L., S.A.B., J.D.G., G.G., A.C.C.V., Y.R. and E.T.; Data curation, A.H., P.S., Y.Y., G.P.M., A.N., J.N.T., G.O. and E.T.; Writing original draft, A.H.; Writing and editing, A.H., D., Y.Y., Y.R. and E.T.; Visualization, A.H., Y.R. and E.T.; Supervision, E.T. All authors have read and agreed to the published version of the manuscript.

**Funding:** This work was funded by the NIST Grant Award Number 70NANB19H024 and by the NSF grant Award Number 1806494.

**Data Availability Statement:** Available via contacting corresponding author.

**Conflicts of Interest:** The authors declare no conflict of interest.

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
