# Peer review of "Background and Blended Spectral Line Reduction in Precision Spectroscopy of EUV and X-ray Transitions in Highly Charged Ions"

_atoms, doi:10.3390/atoms11030048_

Round 1

Reviewer 1 Report

The authors presented the method for background and blended spectral line reduction in precision spectroscopy of EUV and x-ray transitions in highly charged ions. The gave example with extreme ultraviolet spectra of Na-like and Mg-like Os and Ir which were recorded using a grazing incidence spectrometer. The method is based on subtraction applied to time-normalized spectra of highly charged ions of Os and Ir, which degrees of ionization are very similar. Because of that the authors suppose the similar ion charge distributions of heavy highly charged ions that create similar potentials for lighter contaminating background elements and suppose the same plasma condition for residual gases and expect that the subtraction method is justified.

In my opinion the method of subtraction is adequate and very useful in the cases where there are existing of known or unknown residual gases in plasma.

 Minor objection:

The authors mention that there are many important reasons for obtaining precision spectra of highly charged ions due to they being potential candidates for testing of fundamental symmetries and QED because the study of nuclear properties taking advantage of their compressed wave functions.  Also, in general plasmas the states of highly charged ions being the dominant state of matter in the early and current universe.

So that I would like to suggest the authors to emphasize in the text and warn readers to be aware that the method could eliminate background and blended spectral lines originated from lighter ions, but their influence on shift and changes in shape of spectral lines of highly charged ions can not be always neglected, especial in condition of their larger presence and in very high density plasmas. This is necessary to be used in account in precision measurements which could be critical for pushing the limits of QED and the Standard Model.

Author Response

We agree that an additional statement is necessary that explicitly states how this technique reduces background and blended spectral lines since the underlying goal is to reduce the overall systematic uncertainty to below the statistical uncertainty (with no assumption that the systematic uncertainty is eliminated entirely).  Previous systematic uncertainty estimations have been made (Gillaspy, Phys Scr, 2014) to estimate existing shifts. This statement has been added at the end of the paper.

Reviewer 2 Report

I recommend this article to be published following addressing some minor questions. This addresses a good problem in the field and the authors present a solid approach to studying this issue. 

one question I have is around interpreting experiment with CR models. At these intermediate to high-ish electron beam energies the relativistic treatment of inelastic scattering will likely impact the cross sections near the peak of the cross section at 2 to 3 threshold energies. I ask the authors to comment on this observation, and whether the theoretical CR models they allude to are capable of using or calculating relativistic cross sections to account for enhancements to scattering cross section at the peak of the ICS.

Author Response

Since this paper does not address CR modeling, we would prefer not to discuss this issue. However, the atomic collisions code (FAC) can indeed include relativistic effects in ICS.